# Text-Based Games as a Challenging Benchmark for Large Language Models

Qinyue Tan        Ashkan Kazemi        Rada Mihalcea

University of Michigan
{qytan,ashkank,mihalcea}@umich.edu

## Abstract

Text-based games (TBG) are puzzle-solving, interactive dialogue language tasks that have the potential to become a challenging intelligence benchmark for large language models (LLMs). TBGs are similar to interactive dialogue, as they require the capability for bidirectional communication in natural language, while at the same time being straightforward to evaluate in terms of performance, as a score clearly indicates progress in TBGs. We conduct preliminary experiments on FLAN-T5, Turing, and OPT language models to test their puzzle-solving abilities using an *easy* TBG called "Detective". Our results suggest that LLMs underperform in comparison with state-of-the-art and human performance. We discuss the potential reasons behind the performance gap, such as the complexity of turning TBGs into prompts, LLMs not learning from past trials, their lack of memory, and LLMs relying on statistical prediction instead of goal orientation.

## 1 Thinking vs. Language

Large language models have achieved promising performance across many language tasks (Brown et al., 2020; Efrat et al., 2022; Zhang et al., 2023; Srivastava et al., 2022; Goyal et al., 2022), however the extent of their human-like thinking capabilities is still unclear. We argue that text-based games (TBGs) are an important benchmark for the development of Artificial General Intelligence (Laird & VanLent, 2001; Hausknecht et al., 2020), and specifically for improving LLMs. Text-based games, also called interactive fictions, describe the environment and story in natural language and require natural language commands. TBGs can serve as an effective test for measuring the thinking capabilities of LLMs, and high-quality benchmarks such as Jericho (Hausknecht et al., 2020) and TextWorld (Côté et al., 2019) are publicly available to researchers. These games typically involve both language understanding and generation components as well as puzzle-solving elements, making them an ideal testing ground for evaluating a model's performance. The language aspect of the TBGs makes them accessible for LLMs to interact with while the puzzle aspect adds complexity that the models cannot overcome by memorizing training data.

## 2 Text-Based Games and Interactive Dialogue

Generic interactive dialogue is a core NLP task as it closely represents and requires human-level language. TBGs are similar to interactive dialogue in two key aspects. First, the narrative of a TBG is delivered in natural language. Players type text commands to advance in the game, and the game returns textual feedback in response to the player's action. This can be viewed as a dialogue between the player and the game. Second, both tasks require language understanding, memory, and reasoning capabilities. However, TBGs make for easier evaluation than dialogue. A TBG agent interacts with pre-defined games with clear scoring criteria, while a dialog agent may need to interact with real users whose inputs are unpredictable and reliable evaluation metrics are often unavailable.

| Model | Model Size | Avg Score | Max Score |
|---|---|---|---|
| Flan-T5 | 80M | 37 | 50 |
| OPT | 125M | 10 | 10 |
| T-NLG | 7B | 56 | 90 |
| RC-DQN | - | 317 | - |
| Human Performance | - | - | 350 |

Table 1: Scores of LLMs (average and best of 100 trials), SOTA model (average of 100 trials), and human performance (best of 100 trials) in playing Detective. The max score of the game is 360.

## 3 EXPERIMENTS AND DISCUSSION

We formulate TBGs as prompting tasks for LLMs in a similar way to how human players interact with the game. At each step, we input game texts into the models and then texts generated by the models in response to the prompts into the game to perform one step of gameplay. (see Appendix)

We test several LLMs: FLAN-T5 (Raffel et al., 2019; Longpre et al., 2023), OPT (Zhang et al., 2022), and T-NLG (Smith et al., 2022). The text-based game we use is "Detective" (see section 1 in Appendix), an easy game as defined in the Jericho benchmark (Hausknecht et al., 2019). The results are shown in table 3, alongside the state-of-the-art model "RC-DQN" (Guo et al., 2020) which utilizes a deep reinforcement learning (RL) agent, and human performance (best of 100 trials). According to table 3, the scores achieved by large language models are noticeably lower than the SOTA RC-DQN model in playing Detective. By analyzing the LLM outputs, we highlight three main reasons why text-based games are difficult for large language models:

First, TBGs require players to learn from trials that LLMs are not good at. TBGs are designed for players to try many times to solve puzzles. There usually exist multiple actions that seem to be reasonable given the current and previous game texts. Only by trying these options can the player figure out which action is the best one. The connection between the given game texts and the best actions is often not linear. This characteristic makes it hard for LLMs to play TBGs because they are pre-trained and do not support learning from individual users or prompts in real time.

Second, TBGs require players to have long-term memory. Players need to memorize the important information given in the previous game texts to better perform actions in later scenes. However, many LLMs before ChatGPT such as FLAN-T5 and OPT do not contain any explicit memory or state. They rely entirely on the current prompts to take in information. Therefore, when prompts only contain the the game texts at the current step, the models forget about the previous information and have difficulty finding optimal or even reasonable actions. Moreover, simply inputting all the previous information into the prompt is not practical because there are token limits for prompts of these LLMs.

Third, TBGs require players to be aware of their goal that they are playing a game and want to achieve high scores in it. With this goal in mind, players will input reasonable commands that they believe could help achieve their goal. However, LLMs generate text by predicting the next token based on statistical patterns learned from the large training dataset. Therefore, the models sometimes will generate outputs that are reasonable for continuing the prompts while not reasonable for playing a game. For example, the OPT-125M model sometimes ignores that it is a player and instead continues or repeats the scenes described in the game descriptions (see Figure 3 in Appendix).

## 4 DISCUSSION AND CONCLUSION

Our results demonstrate that TBGs are difficult for LLMs. The three reasons: inability to learn from past trials, lack of memory, and not being aware of goal, may be summarized into one, that is lack a way to learn an effective strategy to play a game. We also acknowledge that our experiments are preliminary and do not rule out the existence of a "golden prompt" that effectively solves TBGs using LLMs. Additionally, LLMs evolve and new models such as ChatGPT (Ouyang et al., 2022) exhibit a limited memory of recent interactions.

## 4.1 URM Statement

The authors acknowledge that author Qinyue Tan of this work meets the URM criteria of ICLR 2023 Tiny Papers Track.

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

# A APPENDIX

## A.1 THE DETECTIVE GAME

Below is an example of how the Detective game works in command line.

Figure 1: The Detective game

## A.2 EXPERIMENTAL SETTINGS

We use the TextWorld (Côté et al., 2019) framework to load and play text-based game. We use publicly available APIs to access the large language models. For Turing-NLG, we use the turing API in OpenAI. For OPT and Flan-T5, we use the corresponding APIs in Hugging Face's Transformers. We do not fine-tune the models.

For each model, we play the game for 100 times, each time with a maximum of 1000 steps or until the player is dead. The scores reported in Table 1 are the best and average scores across the 100 trials.

## A.3 DETAILS OF PROMPTS

Prompts into the models mainly come from game texts, with a few words added by us at the beginning or end of the prompts. For the game texts, we use the latest description of game environment instead of the feedback for the player's last action to avoid useless texts such as "This action is not valid". Aside from the game texts, we mainly add two types of information. One is some hints to guide the model about its task of generating actions as a player. The other is some example of valid actions to serve as a few-shot learning.

We give an example of how the prompts and model outputs look like in Figure 2. The black texts are copied from the latest game environment description and the blue texts "What should you do?" are what we added. We limit the number of tokens generated by the model and the responses generated by the models are directly inputted to the game as one step of action. We also give an example of model outputs to illustrate the third reason we discussed in section 4.

Figure 2: An example of our prompt-response pipeline. The black texts in the prompt are copied from game texts and the blue texts "What should you do?" are what we added. Model outputs are sent to the game as one step of action.

Figure 3: An example of when the OPT-125M model ignores that it is a player and instead continues or repeats the scenes described in the game descriptions, as discussed in section 3

