# OpenReview forum: "Text-Based Games as a Challenging Benchmark for Large Language Models"
_ICLR.cc/2023/TinyPapers — Submitted to Tiny Papers @ ICLR 2023_

### Official Review · Reviewer_ZA5w · 2023-03-28

**Confidence:** 4

**Summary Of Contributions:**

This paper proposes to use text-based games to evaluate large language models. They show that LLMs cannot easily solve these games yet.

**Rating:**

High Potential (HP): a submission which meets the reviewing criteria and has potential to make an impact on the field

**Strengths And Weaknesses:**

The authors propose that text-based games can be viewed as an interactive dialogue between the player and the game. It can be used then to evaluate a large language model's language understanding, memory, and reasoning capabilities. RL approaches work best in these scenarios and the authors show that LLMs are not fully capable of grasping the rules of the games and performing reasonable actions.

This is nice contribution to probing scenarios where LLMs fail. Exploring mroe prompting strategies would be insightful to understand how having few-shot examples for instance can help the model to learn the rules of the game.

One main experiment that can show more insight into how good or bad these LLMs are is to check how random selection of one step of gameplay at each time will do. I'm wondering if LLMs are performing even worse than random or not.


**Suggested Changes:**

I would recommend adding a few examples (in the paper or if it doesn't fit in the appendix about how these text-based games look like and how formulating text-based games as prompting tasks is done.

I would also suggest to change the citation style. APA, CHICAGO, HARVARD, or many other styles are available where they also have the author names as well as the year. Having (Smith, et al., 2022) is more informative than just (2022) in the text.

---

### Official Review · Reviewer_ecm2 · 2023-03-30

**Confidence:** 4

**Summary Of Contributions:**

The authors suggest that LLMs underperform on Text-Based Games benchmarks. They report the results on multiple models. They provide discussion points on what the models require in order to perform better.

**Rating:**

Clear, Correct, and Reproducible (CCR): a submission which meets the reviewing criteria

**Strengths And Weaknesses:**

Strengths
1. Authors describe and report experiments in a clear manner.
2. Authors experimented with multiple models and reported scores on the task.
3. Authors provide discussion points on how to develop the LLM-based Text-Based Games model.

Weaknesses
1. There are no details on the experimental setting, including prompts, frameworks, and APIs used. Especially the prompts are missing, which would be directly correlated with performance.
2. There are no comparisons with previous work (RC-DQN) in terms of model difficulties highlighted in the discussion section.
3. The work is only evaluated on Detective TBG.

**Suggested Changes:**

1. Please provide the detailed experimental setting, including prompts and APIs used for LLMs. Generated examples would also clarify the methodology. An appendix could be used.
2. Please also discuss the capabilities and limitations of the previous SOTA model (RC-DQN) and how the LLM-based approach would adopt them.
3. Citation could be provided on how TBG is clearer in terms of evaluation than Dialogue.
4. We could report on one additional task from the benchmarks given spaces around Table 1.

---

### Official Review · Reviewer_zUru · 2023-03-30

**Confidence:** 3

**Summary Of Contributions:**

The authors perform preliminary experiments on FLAN-T5, Turing, and OPT models using an easy TBG called "Detective" and find that LLMs underperform compared to state-of-the-art and human performance. The paper also discusses potential reasons for the observed performance gap.

**Rating:**

Clear, Correct, and Reproducible (CCR): a submission which meets the reviewing criteria

**Strengths And Weaknesses:**

Strengths:
1. The paper presents a clear research question and well-defined findings. The authors evaluate LLMs' puzzle-solving abilities on TBGs, yielding consistent results.
2. The discussion of potential reasons behind the performance gap is insightful and helps the community understand the limitations of current LLMs in the context of TBGs.

Weaknesses:
1. The author didn’t mention if the code will be available.
2. The methodology for converting TBGs into prompts for LLMs is not well-explained. A clear description of this process would help readers understand how the LLMs were tested and how this might impact the result.
3. Including qualitative examples of LLMs' responses and actions in the TBGs would provide a better understanding of the specific limitations and weaknesses of the models.

**Suggested Changes:**

Please see the weakness.

---

### Meta-Review · Area_Chair_CTVq · 2023-04-06

**Recommendation:** Invite to present
**Confidence:** 4

**Metareview:**

Evaluation of LLMs for text-based games, where LLMs underperform on Text-Based Games benchmarks. In addition to reporting results on multiple models, some discussion points on what the models require are also provided.

Very clear aim and well-defined findings. The discussion of potential reasons is well appreciated.

Will the code be made available? Additional methodology explanations and qualitative examples can be added as Appendix. Detailed experimental settings can also be added as Appendix. Double-check the style of in-text references.


**Summary:**

Evaluation of LLMs for text-based games, where LLMs underperform on Text-Based Games benchmarks. In addition to reporting results on multiple models, some discussion points on what the models require are also provided.

**Comments And Feedback To The Authors:**

Please try and make the suggested changes.

**Reason For Not Giving A Higher Recommendation:**

N/A

**Reason For Not Giving A Lower Recommendation:**

Very clear, exciting work. Only very few minor changes are needed.

---

### Decision · Program_Chairs · 2023-04-08

Invite to present